# Biomedical Signal Acquisition Using Sensors under the Paradigm of Parallel Computing

**DOI:** 10.3390/s20236991

**Published:** 2020-12-07

**Authors:** Jesús Jaime Moreno Escobar, Oswaldo Morales Matamoros, Ricardo Tejeida Padilla, Liliana Chanona Hernández, Juan Pablo Francisco Posadas Durán, Ana Karen Pérez Martínez, Ixchel Lina Reyes, Hugo Quintana Espinosa

**Affiliations:** 1Escuela Superior de Ingeniería Mecánica y Eléctrica, Instituto Politécnico Nacional, Ciudad de México 07340, Mexico; omoralesm@ipn.mx (O.M.M.); lchanona@gmail.com (L.C.H); jposadasd@ipn.mx (J.P.F.P.D.); aperezm1410@alumno.ipn.mx (A.K.P.M.); ilinar1600@egresado.ipn.mx (I.L.R.); hquintana@ipn.mx (H.Q.E.); 2Escuela Superior de Turismo, Instituto Politécnico Nacional, Ciudad de México 07630, Mexico; rtejeidap@ipn.mx

**Keywords:** biomedical sensors, biomedical systems, parallel computing, TGAM1, AD8232, wireless systems

## Abstract

There are several pathologies attacking the central nervous system and diverse therapies for each specific disease. These therapies seek as far as possible to minimize or offset the consequences caused by these types of pathologies and disorders in the patient. Therefore, comprehensive neurological care has been performed by neurorehabilitation therapies, to improve the patients’ life quality and facilitating their performance in society. One way to know how the neurorehabilitation therapies contribute to help patients is by measuring changes in their brain activity by means of electroencephalograms (EEG). EEG data-processing applications have been used in neuroscience research to be highly computing- and data-intensive. Our proposal is an integrated system of Electroencephalographic, Electrocardiographic, Bioacoustic, and Digital Image Acquisition Analysis to provide neuroscience experts with tools to estimate the efficiency of a great variety of therapies. The three main axes of this proposal are: parallel or distributed capture, filtering and adaptation of biomedical signals, and synchronization in real epochs of sampling. Thus, the present proposal underlies a general system, whose main objective is to be a wireless benchmark in the field. In this way, this proposal could acquire and give some analysis tools for biomedical signals used for measuring brain interactions when it is stimulated by an external system during therapies, for example. Therefore, this system supports extreme environmental conditions, when necessary, which broadens the spectrum of its applications. In addition, in this proposal sensors could be added or eliminated depending on the needs of the research, generating a wide range of configuration limited by the number of CPU cores, i.e., the more biosensors, the more CPU cores will be required. To validate the proposed integrated system, it is used in a Dolphin-Assisted Therapy in patients with Infantile Cerebral Palsy and Obsessive–Compulsive Disorder, as well as with a neurotypical one. Event synchronization of sample periods helped isolate the same therapy stimulus and allowed it to be analyzed by tools such as the Power Spectrum or the Fractal Geometry.

## 1. Introduction

Performing complex algorithms in less time demands a very high processing capacity, raising drastically the operating frequency, which is unfeasible due to the current limitations of processors such as the size of the transistors, the dissipation of the heat generated or a higher cost [1,2,3].

To deal with this very high processing demand, it is more appropriate to resort to multiprocessing, since it uses different processors where the workload is divided, although it does not increase the working frequency of a processor. Therefore, parallel-computing systems are used both to solve complex problems and to optimize available resources. Derived from parallel-computing systems are distributed computing systems, which in the same way to distribute the processing among different computers, but instead of sharing memory, each computer has an individual one, sharing only the processed information [4].

In a distributed system, computers can be managed in different ways classified mainly as client–server or peer-to-peer architecture. In a client–server architecture, a computer will function as the head of the system or hub node $N_{M}$ and the other computers will be the hosts or secondary nodes $N_{S}$, performing the distributed processing. It is a hierarchy where the highest level is owned by the server (hub) for giving the orders to the hosts and providing the information if required [5].

There are different server-oriented operating systems specialized to work remotely from the cloud or be configured to work from a local area within the same physical space. The Windows Server operating system has a familiar interface for the Windows operating system users. In addition to an operating system, software is required to allow the configuration and execution of code in a distributed manner, e.g., the Matlab R2019b software has a special version for servers to connect computers acting as hosts (workers) and the main computer acting as the hub node from which the programs will be executed; see Figure 1 [6].

The objective of this proposal is to combine a series of biosensors such that in the first place they acquire Electroencephalograms (EEG), Electrocardiograms (ECG), Images from Cameras and Bioacoustic signals. Furthermore, these signals are synchronized in order to generate time series of fluctuations, which are placed at the same sampling moment or epoch. From these signals, time series are analyzed using Power Spectral Density and Fractal Geometry. This proposal can be adapted to indoor and outdoor conditions for helping mental health researchers to verify the interactions among biosensors and to find patterns that interpret the behavior of a patient with a certain mental disorder.

In medicine, distributed systems have been used to capture and process clinical data obtained from different types of sensors for developing or analyzing existing treatments. There are motor rehabilitation treatments that require monitoring of brain stimuli in real time by means of electrodes or other types of sensors. Then, it is necessary to process and analyze the electroencephalograms (EEG) data simultaneously.

EEG data-processing applications have been widely used in neuroscience research due to their high computing- and data-intensity. In the following we present some related works about the usage of parallel-computing systems where EEG neural signals have been captured, processed and/or analyzed.

Yao et al. in [7] construct a parallel and distributed computing platform to analyze the huge EEG data processing with a Beowulf cluster. They study the synchronization measurement of neuronal populations, improving the execution efficiency of EEG data processing by means of parallel and distributed computing at lower-cost.

Gow Jr. et al. in [8] analyze high spatio-temporal resolution intracranial EEG data to understand the functional architecture of spoken language processing. They carry out their analysis on sensor-space broadband activation time series by applying serial versus parallel processing, improving notably the execution efficiency of EEG data processing by using parallel and distributed computing.

Wang et al. in [9] develop the parallel Ensemble Empirical Mode Decomposition (EEMD) to process and analyze EEG neural signals, in order to detect and diagnose several brain disorders. They implement the parallel EEMD with Hadoop in a modern cyberinfrastructure, improving the processing performance, and then increasing the efficiency of the neural signal analysis.

Zhang and Yao in [10] propose parallel convolutional recurrent neural network models for identifying human intended movements and instructions despite incomplete EEG signals. Their experiments on both public dataset and the real-world Brain-Computer Interface (BCI) dataset demonstrate the effectiveness and feasibility of using parallel and distributed computing to analyze diverse human intentions over various EEG resolutions.

Maksimenko et al. in [11] use parallel-computing methods for processing experimental data in real time with respect to multichannel structure of EEG, in order to characterize the interaction between human and machine systems by means of the neurointerfaces BCI. They estimate the spectral properties of multichannel EEG signals, associated with the visual perception. Based on their results, they suggest the possible to detection of specific patterns in multichannel set of EEG data in real time.

Lau-Zhu et al. in [12] carry out a brief overview of recent developments in mobile EEG technologies, claiming that the mobile electroencephalography (mobile EEG) can be a neuroscientific technology to study real-time brain activity, in order to get a deeper-insight about neurodevelopmental disorders from the point of view of non-stationary EEG-based paradigms.

Alsuradi et al. in [13] review emerging literature on electroencephalography, suggesting as a trend the development of neurohaptics systems in the neuroscientific research to understand the complex neural representation caused by tactile and/or kinesthetic stimuli. These neurohaptics systems are being supported by brain-inspired algorithms, computational models of biological neural networks, and biological systems embedded in machines with physical sensing and actuation to model and/or simulate the human sense of touch. The aforementioned topics needs the parallel-computing systems to a better and faster performance.

It is important to highlight that all these works related to EEG neural signals have as their main axis data analysis by means of distributed or parallel processing. Hence, the authors of these works try to find features, patterns or diagnoses related to mental disorders.

Some works can highlight the acquisition or analysis of ECG signals. Qi and Yishan in [14] make an analysis and detection of cardiac arrhythmias in patients between the years 1975 to 1979. They use a database called MIT-BIH Arrhythmia Database, which contains recordings of two ECG channels. This project was programmed in Matlab using a distributed computation of three nodes. Zhang et al. in [15] designs an ECG scheduling algorithm which is a proposal that parallels two nodes to process the information of one ECG channel, it lacks any artificial intelligence tool since it uses the Coffman-Graham algorithm, which is an application of graph theory.

Image acquisition from cameras is an essential task in the field of computer vision, so there are thousands of research articles that capture natural images. However, when it comes to designing local or distributed parallel models the amount of work is drastically reduced. Some works, such as Refs. [16,17], can be considered to be outstanding articles in the field since in both cases they do processing using the Graphics Processing Unit (GPU) using CUDA-Cores, causing the processing of the images to be significantly accelerated, neither of the two works have a stage that synchronizes the acquired images in a serial way, i.e., the non-parallel acquisition with parallel processing. Munoz et al. in [16] do not use any artificial intelligence tool to analyze the captured data, their proposal is limited to measuring the processing speed. In the case of Lee and Yang in [17], SURF key points are used as an artificial intelligence method, making a design without any tool to analyze its results; the authors expose these results in screenshots of their developed system.

There are works that make use of hydrophones mainly for monitoring marine species [18,19]. Both works acquire and process acoustic biosignals outdoors, using geolocation sensors without synchronization stage. Hendricks et al. in [18] use four channels distributed along the Canadian coast, and they use a classifier programmed in Python and show the effectiveness of their results by means of a confusion matrix. And Van Uffelen et al. in [19] program in Matlab programming language three-local nodes of parallel processing, in a buoyancy-driven autonomous underwater vehicle sailing off the coast of the United States; they also present a signal analysis using Power Spectral Density.

The aforementioned related works parallelize the digital signal processing, but do not parallelize the acquisition of EEG neural signals, much less use spatio-temporal synchronization tools to form the most exact sampling times possible. The latter is necessary because brain activity can change in milliseconds, so ensuring the coincidence of the times is transcendental for health experts, doctors or psychologists. That is why the main contribution of this proposal is to obtain simultaneous samples from a large number and variety of biosensors.

On the other hand, physical therapies are applied to treat patients with pathologies and/or disorders. To measure the advances and efficiency of physical therapies, many sensors are required for recording biomedical signal data from the patients undergo these treatments in different situations at the same time, since performing the therapy procedure repeatedly can be stressful or expensive. To collect the biomedical signal data, different sensors are required to be processed by a computer and then data are stored.

Figure 2 shows three EEG neural signals taken in a non-parallel way. These EEG biomedical signals or time series have a delay regarding the others biosignals, and each of these time series belongs to different time lapses or epochs of sampling. Thus, it is shown in a serial capture of biosignals, which prevents the correlation of sensory stimuli between time series, since they occurred at different times [20].

The usage of a computer processor depends on both the amount of information processed and the complexity of the algorithms to be performed in the system. Hence, more sensors, greater the computer computational capacity, leading to more expensive computers [21]. To avoid having many computers processing all the acquired data in a serial way, it is possible to design sets of different $N_{S}$ managed by a $N_{M}$ with little computational capacity, and use a specific processor to be able to emulate the capabilities of several computers. Thus, the present work seeks to wirelessly capture EEG neural signals belonging to the same time span using parallel and distributed computing techniques from a Matlab Server R2019b, and thereby support rehabilitation treatments (e.g., Dolphin-Assisted Therapies) for children with neurodevelopmental problems. It is important to mention that computation speed was neither measured nor compared, since the objective of this work has been simultaneous acquisition due to the assignment of a CPU-Core to each connected sensor. Therefore, in this work the brain activity of both living beings is monitored during a Dolphin-Assisted Therapy, fulfilling the objective of achieving a measurement of brain activity. The present work does not pretend to diagnose any disorder, only to measure and obtain results for its analysis from the engineering point of view but not the medical one.

For this reason, this proposal is further divided into the following sections: In Section 2. Materials and Methods are described the theoretical bases of sensors and the programming of interfaces and execution threads using MatLab R2019b Parallel-Computing System Toolbox. This leads us to propose a system for monitoring biomedical signals using parallel computing as an alternative in the of biomedical signals acquisition. Later, in Section 3. Results, an Integrated Development Environment (IDE) is presented to control the proposed system. The main purpose of this IDE is to perform biomedical signal processing and thereby establish Experimental Results. Then, in Section 4. Discussion, the capture of the samples is analyzed using Spectral Power Density and Fractal Geometry. Finally, in Section 5. Conclusions of this work are presented based on the findings found in this proposal.

## 2. Materials and Methods

### 2.1. Theoretical Basis

#### 2.1.1. Electroencephalographic System: TGAM1

According to Li et al. in [22], electroencephalographic sensors are electronic devices capable of capturing and processing EEG neural signals to later analyze the data obtained for applications in medicine, engineering, neuroscience, entertainment, etc. EEG neural signals can be classified into five different waves or signals:(1)Delta (δ): They oscillate from ½ to 4 Hz, and are generated when the human is in the sleep stage.(2)Theta (θ): They oscillate between 4 and 8 Hz, and appear during meditation, tension or frustration.(3)Alpha (α): They are found from 8 to 12 Hz, and are related to a mental state of relaxation.(4)Beta (β): They range from 12 to 30 Hz, and are present during concentration state or when a mathematical problem is being solved.(5)Gamma (γ): They usually oscillate around 40 Hz, and are related to perception.

Figure 3 shows the device used as the interface between the brain and the computer called the ThinkGear ASIC Module 1 (TGAM1). TGAM1 is a non-invasive method with a reliability degree of 98%. TGAM1 module is connected directly to a dry electrode, it has an EEG channel with three contacts: EEG, REF and GND. Moreover, it has an incorrect setting signaling that detects through the “Poor Signal” warning if communication with the user’s head should be reestablished, and it is also provided with advanced filtering technology with high immunity to noise [23]. Its low power consumption is suitable for portable battery-powered applications, since its maximum power consumption is 15 mA at 3.3 V.

On the other hand, a Bluetooth connection is used for data transmission to 57600 bauds [24]. Therefore, in this work the HC-06 Bluetooth module is used, especially the B26782H with the following features: USB protocol 2.0, Bluetooth Protocol Version 2.1 + EDB, Range about 10 m, Class: 2, ISM band from 2.40 to 2.48 GHz, and Security by means of Authentication and Encryption, Guangzhou HC Information Technology Co., Ltd., Guangzhou, China. In Figure 3 is shown the B26782H Bluetooth module, where it can be seen the microprocessor, oscillator, the antenna integrated in the PCB, connection ports and resistors and filtering capacitors [25].

#### 2.1.2. Electrocardiographic System: AD8232

The electrocardiogram (ECG) is applied to examine the electrical function of the heart. ECG manages to distribute data from activity in heart. Analysis makes possible to have information about the cardiac balance, the volume, the myocardium and the activity of the heart chambers. The ECG uses small electrodes placed in defined places: arms, chest and legs [26].

To measure ECG signals, our proposal uses the ECG model AD8232 to facilitate the recognition of the electrical function of the heart by means of a non-invasive procedure in which the cardiac function and rhythm are estimated by recording the electrical function of the heart. In the electrical activity, saving is achieved by placing electrodes on the chest, arms and legs, Figure 4. The AD8232 component has filter and signal noise amplifiers, particularly tuned for ECG signals. This model cancels out 60 Hz noise. The module has an analog type output, it is only necessary to solder the pins and connect to a microcontroller such as the Arduino [27].

The ECG sensor is an electronic circuit corresponding to the part coming into first contact with the measurement signal after its capture, performing the appropriate basic processing of this signal to later treating it satisfactorily. Typically, an ECG sensor includes the tasks of amplifying and reducing certain noise components. One of the most used ECG sensors is the AD8232 due to its capacity for acquiring the cardiac signal and converting it into a digital signal, and then communication with Arduino can be carried out [28].

Thus, the AD8232 ECG sensor is designed to extract, amplify and filter small biopotential signals in the presence of noisy conditions such as those obtained in the placement of distant and moving electrodes. The AD8232 heart rate monitor has nine connections. The five pins needed are labeled GND, 3.3 V, OUT, LO-, and LO+. On the one hand, it has a low power supply, allowing it to be powered by the ESP8266 (NodeMCU) development board, which is going to be used without supplying power problems for it. The amplification of this ECG sensor is high and allows the enhancement of the signal, but also the noise can be filtered later to eliminate its effects [29].

#### 2.1.3. Bioacoustic Wave Acquisition System: Bruel & Kjær 8103

To measurement bioacoustic signals, a hydrophone with a high sensitivity Data Acquisition card (DAQ) is used. Low-frequency underwater hydrophones are used in conditions greater than 100 m in open sea, and operate below 500 KHz. Conventional hydrophones measure no more than 9 cm and have a diameter varying between 10 and 25 mm, and they have a polyurethane encapsulation for an attenuation in the frequency and a faster sensitivity. In this work, a Bruel & Kjær Hydrophone Type 8103 (Figure 5) is used to study marine life because of its miniature size, durability, resistant to corrosion, ability of calibrating reference standards and capacity of ultrasonic measurements in liquids or gases [30].

Figure 6 shows the polar pattern of the Bruel & Kjær Hydrophone Type 8103 for picking up sound from different directions, canceling out unwanted angles [31]. Also, a Bruel & Kjær Type 3670 Data Acquisition card is used as an accurate and reliable DAQ hardware unit to measure sound and vibration on electroacoustic products. This DAQ has an 8-channel input by 2-channel output serial port, allowing communication with almost any high-level, script-oriented programming language such as MatLab or Python. Hence, it was possible to develop graphical user interfaces, data analysis and mechanization of mathematical procedures. Figure 7 shows the DAQ Bruel & Kjær Type 3670.

#### 2.1.4. Image Acquisition System: Logitech C920

Feng et al. in [32] establish that an image is captured by a device capable of measuring waves located in a certain region of the electromagnetic spectrum and obtaining a useful representation of a given scene. The natural example is the human eye, perceiving wavelengths from 400 to 800 Hz, and other examples are pinhole cameras and cameras with Charge-Coupled Device (CCD) or CMOS (Complementary Metal-Oxide Semiconductor) sensors; see Figure 8a [33].

This proposal makes an image capture by using a specific CCD sensor of a Logitech C920 camera (Figure 8b), whose maximum resolution is 1080p/30 fps and 720p/30 fps. Its focus type is automatic and comes with an integrated stereo microphone used as an ambient sound-capture card and a universal clip compatible with tripods for monitors, screens or laptops [34].

#### 2.1.5. Parallel-Computing System: MatLab R2019b

Matlab is a *high-level* programming language and an interactive technical-scientific computing environment. It includes functions for the development of algorithms, data analysis, numerical calculation and visualization. Using Matlab, technical computing problems can be solved more quickly than traditional programming languages (e.g., C, C ++, or Fortran). In sequential programming, an instruction is not executed until the previous one has ended. However, in parallel computing many instructions run simultaneously. In November 2004, MathWorks released two MATLAB tools called *Distributed Computing Toolbox* and *Distributed Computing Server*, becoming later *Parallel-Computing Toolbox* and the Distributed Computing Server, respectively [3].

The classification of Parallel paradigm in some computers is directly related to the level of parallel tasks that their hardware can support: multicore and multithreaded computers have multiple elements of processing on a single machine, while clusters or distributed computing employ multiple computers to work on the same task [35].

On the one hand, the Parallel-Computing Toolbox offers different parallelization tools depending on the needs, either to perform faster processing or because the data is too large to enter memory. On the other hand, the MATLAB Distributed Computing Server is made up of several workers receiving computing tasks from the client side through the functions implemented in the Parallel-Computing Toolbox (Figure 1). Thus, a worker is a process running on a Distributed Computing Server cluster.

*Parallel-Computing Toolbox* allows running programs with a high computational load using multicore processors, Graphics Processing Unit (GPU) and clusters. This toolbox makes use of the processing capacity of the computer hardware, executing the applications in several workers working locally [36]. The applications can also be run, without modifying the code, in a cluster or in a grid computing service. Therefore, *Parallel-Computing Toolbox* can improve performance in several situations:Applications with repetitive code segments and loops. Each iteration is evaluated separately in a parallel loop with the only restriction that these repetitions must be independent of each other.Programs with a series of tasks that do not depend on each other. A parallel loop can also be implemented.Evaluation of the same code on different sets of data at the same time. To do this, a set of workers is used for working at the same time with the same code, but with different data.Information is too large to be stored in the computer or server memory. Therefore, it must be distributed among different workers in such a way that each one works with a part of the data.Performance improvement if the code runs in parallel or on a GPU.

Figure 1 shows the Matlab architecture *Parallel-Computing toolbox*, allowing a Matlab client to work with several Matlab sessions at the same time. The *Parallel-Computing Toolbox* allows working with all workers or CPU Cores on a local machine, limiting this to the number of cores and threads of the microprocessor (in our case, four). If there is an application requiring more than four CPU cores in a cluster, the Matlab Distributed Computing Sever toolbox must be used, allowing working with all those workers or Distributed CPU Cores [37].

### 2.2. System for Monitoring Biomedical Signals Using Parallel Computing

This proposal is designed to assist children with neurodevelopmental problems, but it is possible to use the present system to help any health professional to investigate several disorders or pathologies that require biomedical signal processing. Moreover, the system can be adapted to the specific needs of the research area, since it can grow or be modified and it is only limited to the capabilities of the local microprocessor. Specific sensors were designed to study what happens to a patient during a Dolphin-Assisted Therapy (DAT), but this work does not give conclusive information if it affects a patient in a positive way, for this it can be referred to [38,39]. To validate the proposed integrated system, it is used in a DAT in patients with Infantile Cerebral Palsy and Obsessive–Compulsive Disorder, as well as with a neurotypical one. A DAT is the interaction between a bottlenose dolphin (*tursiops truncatus*) with humans, affecting presumably in a positive way to the patient central nervous system by means of the dolphin echolocation.

#### 2.2.1. Architecture

Figure 9 shows the architecture of this proposal, consisting of the acquisition of the following signals:EEG electrode attached to a patient with neurodevelopmental problems.EEG electrode attached to a bottlenose dolphin (*tursiops truncatus*).ECG electrode attached to a patient with neurodevelopmental problems.USB-DAQ Bruel & Kjær Type 3760 with an 8103-hydrophone (introduced into a tank with salt water).Logitech C920 USB Webcam for a front view of the DAT.Logitech C920 USB Webcam for a side view of the DAT.

The microprocessor used is the Intel Core i5 4570, with four CPU Cores and up to eight threads of execution can be carried out, belonging to the DELL Optiplex model 9020 system. Moreover, the electrodes (*BIOSIGNALS* or $EEG_{M}$, $EEG_{S}$, and $ECG_{S}$ ) are specially designed for the application during a DAT taking into account the current Official Mexican Standard NOM-135-SEMARNAT-2004, while the USB signal acquisition devices (*ADC-SIGNALS* or USB-DAQ${}_{S}$, USB-Cam1${}_{S}$ and USB-Cam2${}_{S}$) are configured with the given configuration by the manufacturer. According to Figure 9, the three *BIOSIGNALS* use each one a CPU-Core exclusively due to the nature of their precision, while the *ADC-SIGNALS* use a single CPU-core in a multiplexed manner.

#### 2.2.2. Design of Biomedical Signal Interfaces

A biosensor is a device that transforms the energy of some analog magnitude into voltage or current values of a living being. Electrical biosensors produce a voltage and/or a variation of resistance as an output used to characterize to the sensor; this voltage can be quantified, measured and later used to control a system. These biosensors are generally made up of discrete components such as capacitors, resistors, coils, and diodes. Our design of the biomedical signal interfaces lies mainly in the connection and configuration of the EEG (Section 2.1.1) and ECG (Section 2.1.2) sensors, and these devices must be completely waterproof and meet the following characteristics:Salinity: 18 to 36 parts per thousand.Hydrogen Potential (pH): between 6 and 8 units.Temperature: from 5 to $27{\;}^{\circ }$C.Pressure: 1 Atmosphere (ATM).

Even these EEG ($EEG_{M}$ and $EEG_{S}$) and ECG ($ECG_{S}$) cases support at least one atmosphere of pressure, i.e., they do not leak at 10 m of depth, they were designed for floating.

The design of TGAM1 electroencephalographic devices made it possible to measure the brain activity of both a patient and a female bottlenose dolphin during a DAT, generating an electroencephalogram (EEG) to help experts understand what happened during DAT. Figure 10a shows that two types of EEG interfaces designed in this work:(1)$EEG_{M}$ for a patient was placed at the $F_{p1}$ or Frontopolar 1 electrode, according to the 10–20 system; see Figure 10b.(2)$EEG_{S}$ for a female bottlenose dolphin was placed at a longitudinal distance of 6 cm from the blowhole; see Figure 10c.

The $EEG_{M}$ and $EEG_{S}$ brain activity monitoring systems record samples from brain activity during a DAT, for a specific time, since the dolphin mainly requires constant submersion. Therefore, the measurements are carried out during periods of five minutes maximum because the device is not designed to perform measurement under salt water due to the loss of connection and attenuation of the signal captured by the electrodes.

In general, both $EEG_{M}$ and $EEG_{S}$ consist of a ThinkGear ASIC Module electroencephalographic sensor -responsible for collecting raw data on the brain activity of both living beings, two reference electrodes and a main electrode. All of them are the means by which the TGAM1 sensor receives the samples, and a Bluetooth module sends the collected samples to a CPU-Core to later be stored in a text file with a *.txt* extension; see Figure 10a.

To develop the program used to read the data from the sensors, we apply the development tools provided by the manufacturer NeuroSky, including licenses, communication protocols, libraries, among others. These libraries contain the necessary code to extract the data received when there is physical contact between the device and the frontal lobe of the patient or dolphin.

On the one hand, $EEG_{M}$ Electrode adaptation for a patient consists of keeping the headband but introducing the TGAM1 sensor along with Bluetooth transmitter in a waterproof case purposefully designed, Figure 11 shows the Modeling (a) and Printing (b) of this case. For 3D printing, the parameters of the water quality in the tanks are considered. For $EEG_{s}$, the points where it is possible to record brain activity from female dolphin are measured; see Figure 12a. In addition, a comfortable and easy-to-handle holding device is proposed; see Figure 12b.

On the other hand, $ECG_{s}$ consists of the AD8232 electrocardiogram sensor, used to collect data on the cardiac activity of the patients obtained from the three electrodes distributed on the chest. A Bluetooth module sends the collected samples to a computer to be stored in a *.txt* file. The mica to protect the conductive gel is detached, and the three electrodes used are placed forming a triangle (Figure 13a) mainly on the patient’s thorax, looking for an area of the skin as flat as possible, avoiding bony areas.

To develop the program used to read the data from the sensor, the Arduino environment is adapted to use the *NodeMCU* module, *ESP8266*. The analog signal is converted into a digital signal by means of the *NodeMCU* board. Moreover, the data acquisition is based on the reading of $ECG_{s}$, collecting the potential of the electrical signal of the heart, and then the signal is converted into digital to be transmitted by Bluetooth.

The program developed starts with the *SoftwareSerial* library declaration, since this allows the *Atmega* chip to receive serial communication during the time it works on other tasks; this library also allows us to use any available pin on the card as a serial port. A new serial object *btserial* is created, using some pins 4 of *NodeMCU* board to work *Tx* and *Rx* for Bluetooth.

To avoid any accident, we consider a waterproof structure, easy to handle, and strong enough for safe and reliable handling. Hence, a *Sonoff*-like box (Figure 13b) is chosen with transparent cover material, 132.2 mm × 68.7 mm × 50.1 mm dimensions and a weight of 145.0 g.

#### 2.2.3. ADC-SIGNALS Configuration

As mentioned in Section 2.2.1, the ADC-SIGNALS interfaces are configured with the parameters and drivers provided by manufacturers without physical or logical reconfiguration. The ADC-SIGNALS are connected to the same CPU-Core in a multiplexed since the Bruel & Kjær sound card type 3760 has a start-stop flag. Therefore, between the first and the last sample, images are taken alternately by devices USB-Cam1${}_{S}$ and USB-Cam2${}_{S}$.

USB-DAQ${}_{S}$ is configured using the Matlab R2019b *audioDeviceReader* function and an *ASIO* driver is established. Additionally, the following characteristics are established: Number of Channels 1; Sampling Frequency 96,000; Samples per Frame $2^{17}$, i.e., 131,072 samples at 128 kS; Sensitivity of Hydrophone 8103 in Volts = $25.41\times 10^{-6}$ V/Pa; Sensitivity of Hydrophone 8103 in Pascals 0.0944 pC/Pa; Sensitivity of 2647A coupler 0.001 V/pC; and BitDepth of 24 bits, Bruel & Kjær, Nærum, Denmark.

For the configuration of the webcams USB-Cam1${}_{S}$ and USB-Cam2${}_{S}$, the video sockets are first created with the following instructions:



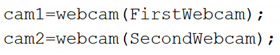



A VGA resolution is set for the embedded capture as follows:



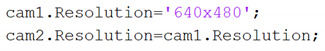



And then, a multiplexed access is made using the following instruction:



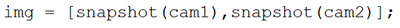



The results of both captures are saved in a single concatenated image in a Tagged Image File Format *.tiff*, to compensate for the quality losses due to the established VGA resolution.

#### 2.2.4. General Algorithm

Like all programming languages, Matlab has an instruction set for iterative or decision instructions such as *if-else*, *switch-case*, *do-while*, *for*, etc., so a *for* loop has the following form:



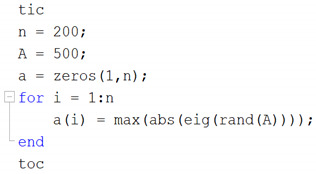



The execution time of this loop is 31.94 s, which is weak for the high-level languages based on scripts with precompiled code such as Matlab. Hence, to speed up the execution time from its R2008a version, a parallel-computing instruction called *parfor* is introduced, looking similar to *for*, thus the previous example would be as follows:



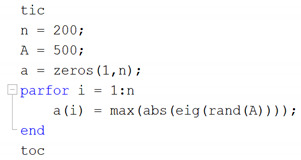



This example is completed in 10.76 s and used all the resources available to Matlab, accelerating the same procedure three times. *parfor* can accelerate processes by assigning a task from index *i* to a CPU-Core. For four workers, four-simultaneous iterations can be performed, at the end of the *parfor* cycle all instructions will be made but the order will never be in Ascending order since the response of each CPU-Core is different. Therefore, one CPU-Core can do more than one instruction while another does only one, depending on the particular load of each CPU-Core. Thus, the content of the iteration must be mutually exclusive or totally independent of the previous instruction.

Most of the applications reported in the literature use *parfor* to accelerate the processing time. This proposal uses it to have controlled access to each CPU-Core, so the speed of the CPU-Core is not so important but the quantity. Hence, all the resources of the Intel i5 4570 microprocessor are configured as follows:



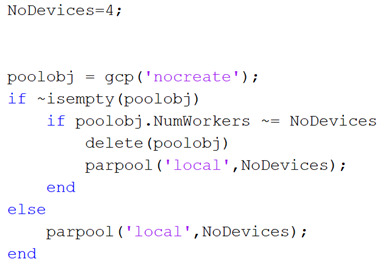



Therefore, the index *i* has a value with the total available workers and by means of a *switch* instruction it evaluates in an orderly way the access and the number of the CPU-Core, as follows:



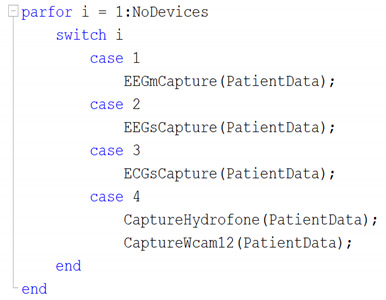



We summarize the proposed General Algorithm of this system by means of Figure 14, due to the huge amount of code developed, which is divided into many functions and subroutines. In this way, first the existence of a *TXT Flag File* is verified; if this file exists, it is deleted and the κ index is initialized with the number of necessary samples or Δ. Then, all the devices acquire their respective time series, acquiring infinitely. $EEG_{M}$ controls κ and creates *TXT Flag File* when the counter reaches zero. When all devices, except $EEG_{M}$, detect the existence of the *TXT Flag File*, they stop their acquisition. For the BIOSIGNALS interfaces, a stop-band filtering is performed to remove the noise induced by the 60 Hz power supply line, allowing use of a Notch frequency of $60/(F_{s}/2)$ for 3 dB of bandwidth, where $F_{s}=512$ for $EEG_{M}$ and $EEG_{S}$; and $F_{s}=8000$ for $ECG_{S}$.

Finally, all the acquired time series pass-by a time rectification or synchronization process, explained in detail in Section 2.2.5.

#### 2.2.5. Synchronization

A time located event in such as a visual stimulus, an interaction in a therapy or an acoustic signal are defined as Epoch, in the field of neuroscience. In the same way, the procedure to extract continuous biomedical signals from a specific time-window is known as biomedical signals Epoching. That is why it is critical to sample sensor interactions at the same sampling time. There are several techniques for using a sensor network, emerging as solutions for combining optimally multiple biomedical signal sampling sources. One of these techniques widely described in the literature is Distributed Data Fusion (DDF), consisting of a network of process nodes connected to each other to share a common goal [40,41,42]. The latter fact is the main difference between the DDF and the present proposal since there is a single CPU-Core that directs the acquisition of time series, using its own internal clock and it is not necessary to establish communication with other nodes. However, in systems that use the DDF, the sensor data is acquired via multiple nodes and communication between these nodes is the biggest bottleneck. Our proposed system tried to avoid this bottleneck for ensuring the temporality of heterogeneous samples from different biomedical signal sensors. Hence, the monitoring of biomedical signals does not improve the quality of the samples, this is done by filters that reject a certain frequency band; this synchronization procedure rather ensures the same sampling time to a stimulus applied to the patient.

According to Figure 14, all the acquired time series need a time rectification or synchronization process since all processes are asynchronous each other, only controlled by the *TXT Flag File*. Together with the sample taking with a discretized value of voltage, pressure or image, each of these samples is labeled with its respective Unix time number in POSIX time format or $U_{time}$. The latter is the number of seconds with passed since 1 January 1970, and it is a way for the processors to have the same time base; see Figure 15.

The precision for the present project is microseconds because the Hydrophone Bruel & Kjær 8103 has a frequency sampling rate $F_{s}$ = 96,000.

As shown in Figure 16, the first sample of each time series acquired by the devices connected in parallel is identified as $S_{1}$, and *t* is highly unlikely that all first samples would have the same $U_{time}$ label. Hence, the highest $U_{time}$ at this Unix timestamp is searched for in order to consider the lower cut-off threshold or $\mu_{b}$ of all the time series, and then all the time series would start at the sample $\vec{S_{1}}$. In the same way, the label of all the latest samples or $S_{i}$ is found and the lowest is verified, finding the lower threshold or $\mu_{t}$, and thus ensuring all the time series end up in the $S_{\epsilon }$ sample.

Despite performing the process described above, it is not ensured that the length of the time series is the same since there are different sampling rates, but what is ensured is that they all comprise the same period in real time, ensuring the same sampling epoch; see Figure 17. Given the presence of many parallel-computing instructions, in terms of the function test, we performed six tests in order to get more accurate results, which were developed with one of the team members, who we considered to be a patient with a neurotypical brain.

## 3. Results

### 3.1. Integrated Development Environment

Using the Matlab GUIDE tool, an Integrated Development Environment (IDE) is developed; see Figure 18. It consists mainly of three windows:(1)System Configuration (Figure 18a),(2)Patient Information Capture (Figure 18b), and(3)The Case Study (Figure 19).

In the System Configuration Window, the reconfiguring of the system according to the researcher’s needs is possible by adding or removing parts of the system. Figure 18a shows the use of the system. All devices must be recognized by the computer via connection-verification buttons. In the case of bioacoustic signals, first they must be calibrated and verify if the ambient noise can affect the sample collection. Thus, a pure tone of 5 KHz is sent, and using the Welch Power Spectrum it is evaluated if this tone is the predominant signal; if so, the Bruel & Kjær Hydrophone 8103 can be considered to be calibrated.

Once the researcher configures the system with the parts considered necessary, the patient data is captured, such as her/his name, diagnosis, sample duration and the moment of taking the sample; see Figure 18b. Researchers in the neurosciences develop experiments before, during or after a certain therapy. Nonetheless, the present work is used in an Dolphin-Assisted Therapy but it is not limited to DAT, on the contrary they can be motor or speech therapies, measuring brain activity, depending on the stimuli offered by researcher.

Therefore, with the system configuration and the patient data, respective time series are captured in parallel, then filtered from additive noise and synchronized. Once the system is finished, the *Take a Sample* button must be activated to capture more samples, if necessary.

### 3.2. Signal Preprocessing

Once the samples are recorded in situ with the patient and under the necessary conditions, a researcher must be provided with analysis tools. That is why the *Analysis of the results of a patient* window was designed; see Figure 19. Using the *Search Patient* button, folder containing results is searched and the basic information and the devices that took the samples from the case study are displayed. Thus, the configuration of the system with which the study case samples were captured is also shown, in particular for this example the following configuration was used in the capture: $EEG_{M}$, USB-Cam1${}_{S}$, USB-Cam2${}_{S}$, and USB-DAQ${}_{S}$.

From Figure 20, depending on what biomedical signal will be analyzed, the system will display Time Series, Periogram, and General and Average Spectrograms based on the Welch power spectrum. If it is a Time Series from the $EEG_{M}$ or $EEG_{S}$, the average power spectrum is divided into the characteristic frequencies of the EEGs, i.e., δ, θ, α, β and γ, while if the data are acquired by $ECG_{S}$ or USB-DAQ${}_{S}$, the data are presented in octaves. Also, this figure shows the representation of the Raw Time Series (Top-Left), the Periodogram pointing out how much power has a given frequency (top-right), the Spectrogram showing the relationship of how much power has a frequency at certain time (bottom-left), and the Average Spectrogram indicating a band of frequencies that has a certain average power during the sampling period (bottom-right).

In the case of the cameras USB-Cam1${}_{S}$ and USB-Cam2${}_{S}$, no digital processing of these signals is performed, nor are patterns found, only a video is made with one or both shots together with the chosen time series results.

### 3.3. Experimental Results

Our experimental results are yielded as follows. The *System for monitoring biomedical signals using parallel computing* is applied in a dolphin-assisted therapy (DAT) of a patient with neurodevelopmental problems, specifically spastic infantile cerebral palsy. For testing the $EEG_{M}$ device, two electroencephalographic captures are made before DAT (Figure 21a) and during DAT (Figure 21b), while (Figure 21c) shows the time series generated. A HC-06 Bluetooth Module is used with the IEEE 802.15.1 v. 2.0 communication protocol. Thus, in theory there are 100 m of connection range outdoors. This range decreases indoors until a connection allows permanent communication at approximately 10 m. In an experimental way under salt water, the standard range is reduced by 97%, i.e., a coherent connection and transmission are obtained at a maximum of 3 m because the patient is on a platform submerged in a saline solution, increasing conductivity and energy going to the ground directly [43], Guangzhou HC Information Technology Co., Ltd., Guangzhou, China.

In the case of the $EEG_{S}$, previously the dolphin is trained to allow the sample to be taken (Figure 22a), then it was verified if the TGAM1 sensor keeps continuous connectivity. This is done by means of a flag called *POOR SIGNAL*. Figure 22b shows the connection values between the device and the dolphin:Values equal to 200 indicate no connection, so the data recorded by the sensor is noise.Values < 51 point out that the recorded data are consistent.Values equal to 0 mean an optimal connection.

According to Figure 22b, after a sensor-initialization period there is a connection with enough strength to be considered coherent, and then the time series of Figure 22c is obtained.

For the $ECG_{S}$ device, the gel patches formed in a triangle way are placed on the patient’s chest, obtaining the time series of Figure 23. The device-sampling rate setting $F_{s}=1000$ samples per second, establishing the connection between the COM and Bluetooth communication ports. In this figure, we can also see the characteristic peaks and valleys of an electrocardiogram to be used for studying the heart rhythms and defining the patient mood, at the same time that their hearth activity is known. For the development of the program, a Matalab Mexfile is used for interoperability between the Python programming language as it is multiplatform and multiparadigm. Once this bridge application is created between both languages, it is necessary to include libraries such as Serial, allowing communication with the board via the serial port. *COM10* is defined as the output port for data transmission between the Bluetooth with the PC; the following parameters are configured as follows:*baudrate* is the speed that is transmitting, 115,200 baud.*bytesize* is the size of data, 8 bits.*parity* is the parity of an error checking way, used in serial communication.*stopbits* are the stop bits to signal the end of communication.

The present work carries out an analysis of the frequencies produced by the dolphin, thus the environmental noise is attenuated and the software tools necessary to the visualization and analysis of the frequencies are developed. Hence, the fundamental tool used for this work is the Bruel & Kjær miniature Hydrophone model 8103, manufactured for industrial and marine fauna research purposes, with a high frequency response, capable of measuring in a range from 0.1 Hz to 180 KHz. Figure 24 shows the recording captured by the USB-DAQ${}_{S}$ device with a high sensitivity; this sample contains the pressure of a bioacoustic signal to Bruel & Kjær Hydrophone 8103, and it can be seen general disturbances and attenuation treated digitally.

Finally, Figure 25 shows an image taken by devices USB-Cam1${}_{S}$ and USB-Cam2${}_{S}$ during a DAT. The same instant of the taking is observed in two views: (i) in front of the patient (left) and (ii) from the side to the DAT (right). Depending on the configuration and researcher needs, both cameras or just one can be used. The sampling rate is 30 pairs per second.

## 4. Discussion

*System for monitoring biomedical signals using parallel computing* is a multipurpose tool for the study of neurosciences, so this section exposes certain technical details for making this proposal an alternative for researchers.

The purpose of biomedical signals monitoring, in general, represents a process of acquiring, processing and either automatic or manual classification of the results. The present proposal has certain parameters useful to psychological or medical researchers to determine pathologies in patients with problems in the central nervous system, resulting in neurodevelopmental problems. Therefore, two types of tools for digital analysis of signals obtained by the system are proposed to support medical and psychological decisions about patients: (i) Welch Power Spectrum Density and (ii) Self-Affine Analysis.

On the one hand, an analysis of biomedical signal interfaces captured after being filtered by a Notch filter shows how the noise of the electrical power line affects both the samples of the $EEG_{M}$ and $EEG_{S}$ electroencephalographic devices (Figure 26a,b, respectively), as well as the electrocardiographic signal from the $ECG_{S}$ device (Figure 26c). For this reason, it can be claimed that a power signal of 110 V/60 Hz significantly affects the acquisition of any biological signal. To obtain a normalized error of 1000/60, a 3 dB Kaiser window is implemented. Furthermore, Figure 26 shows time windows of 0.4883 and 0.25 s for the EEG and ECG sensors, respectively. These window sizes were chosen for illustrative purposes and thereby to show the effects of additive noise from a 60 Hz power line.

Figure 27a,b show an analysis of the Welch Power Spectrum using a Periodogram to the data acquired by devices $EEG_{M}$ and $EEG_{S}$, respectively. This analysis indicates that the power at certain frequencies (at certain time) only takes into account the periods where the connection is stable. Moreover, studying the activity frequency versus power measured in dB only for the samples captured between frequencies from 0 to 60 Hz, minimal samples during disconnection periods are observed, considering in general the signal as valid. For the $ECG_{S}$ Device, Figure 27c shows a bandwidth up to 500 Hz; this Spectrogram (in decibels referred to Hertz) indicates that the frequency reached more power. In addition, we can see that the highest power is reached by the frequency of 1 Hz, meaning a patient’s heart rate on average of 60 bpm (beats per minute). So to this electrocardiographic system it can be added parameters to enhance the ECG interpretability such as heart rhythm variability measures, for instance.

Figure 28a,b show the representation in an Average Spectrogram of the brain activity from the $EEG_{M}$ and $EEG_{S}$ devices, respectively. From a medical or psychological point of view, in both electroencephalograms the β-band is predominant, pointing out an apparent concentration state in both the patient and the dolphin. For the patients, an important change is seen in the attention behavior, while for the dolphins it could be due to familiarization with the $EEG_{S}$ device; perhaps the animal concentrates, waiting to receive the order from the coach. Figure 28c shows the Average Spectrogram with a sampling rate of $F_{s}=1000$ samples per second; in octaves 3 and 4 a greater power in the band is displayed, so the predominant average heart rate of the entire time series ranges from 4 to 16 Hz. In the octaves 11 and 12 no power is obtained because the frequency is about 1024 Hz, exceeding the maximum sampling rate or $F_{s}/2$, according to Nyquist–Shannon sampling theorem.

Figure 29 presents a Spectrogram of time versus frequency of the data obtained by the USB-DAQ${}_{S}$ device, showing that noise appears at certain periods of the sample. But the most important, in almost all the time a frequency of 13 KHz appears, corresponding to the *clicks* emitted by dolphins when they use their echolocation [44].

Therefore, this power spectral analysis provides tools in the frequency domain to the neuroscience researcher to study the behavior of certain biomedical signals, such as EEG, ECG or bioacoustic signals. When it is necessary to develop a deep study in the time domain, an analysis of the average fluctuations over time series can be implemented using tools from Fractal Geometry such as Self-Affine Analysis.

Fractals are set of shapes normally generated by a process of repetition or iteration, with full-scale detail and infinite length, by not being differentials and whose dimension is non-integer or fractal [45]. Fractals are self-similar, implying that any portion of them, if scaled up, would have an identical appearance to the whole object. However, the most objects in nature are not self-similar, without the scaling characteristics occurring in only one direction, so we have self-affine fractal objects. The appropriate scaling to keep invariable the self-affine aspect is called self-affinity.

Self-Affinity is a mathematical tool used by us to estimate whether there are correlations in time series. Positive correlations indicate if there is a rising tendency, it will continue rising over time, while if there is a falling tendency, it will continue falling. The scaling exponent of Hurst (*H*) points out what kind of correlations are exhibited by time series.

If 0.5≤H<1, time series display positive correlation (or persistent behavior) fitted by a Power Law. If 0≤H<0.5, time series exhibit negative correlations (or antiperistent behavior) fitted by a Power Law. Lastly, if H=0.5, times series display no correlations, or random walk behavior.

Figure 30 shows the Self-Affine analysis carried out in this work. In the case of the signals from sensors $EEG_{M}$ (Figure 30a), $EEG_{S}$ (Figure 30b) and $ECG_{S}$ (Figure 30c), they display a persistent behavior since all the values of the Hurst exponent *H* are greater than 0.5, fitted by a Power Law (Straight Line).

In Figure 30 there are shown 200 long-term fluctuations. Every 200 samples the standard deviation of the subsample or sampling window $\delta \left(t\right)$ is determined, moving by one throughout the time series. Therefore, time windows are generated throughout the sample and these deviations are averaged [circular marks (∘)] on the graph, giving the variability.

Hence, in Figure 30a–c, once the $\delta \left(t\right)$-value increases, the σ-value also increases (see the Power Function Fit line), since the relationship between $\delta \left(t\right)$ and σ is linear, a positive correlation is displayed. Therefore, each time the fluctuations increase, the dispersion is more stable with respect to the correlation and, as a result, long-term stability is obtained, pointing out both that the sample was taken correctly and that the brain of patient or dolphin have assimilated the environmental information of the initial transitional period, which indicates that the therapy is working in a an initial period of time and then stabilize its long-term efficiency.

Finally, it is important to compare the present proposal with the existing works. Table 1 shows the main characteristics such as the type and quantity of biosensors, if a synchronization stage is taken into account, the artificial intelligence tools applied, as well as the programming language employed to develop the algorithms, with the aim of comparing a certain number of systems and their features with state-of-art works. According to Table 1, our proposal (in bold) is a design that encompasses acquisition and analysis tools and is the only one that combines the sensors most used in neurosciences and health sciences, because it uses the largest amount of CPU Cores simultaneously. The Development in Matlab of this work enhances its future applications since it can process data rather in locally or distributed way. We can add cascading classifiers or neural networks to find patterns in the images. Also, our proposal can be extremely resistant to heat, water, humidity or salinity. The systems aforementioned in the state-of-the-art do not isolate the periods of cardiac and cerebral activity of a patient. In our proposed system the acoustic environment where the events occur can be visualized by making a video analysis of interactive behavior of both the animal and the patient, allowing psychologists to do studies of animal behavior.

## 5. Conclusions

Dolphins are very complex animals, since these cetaceans have advanced communication like human beings do. Therefore, in this work the brain activity of both living beings is monitored during a Dolphin-Assisted Therapy, fulfilling the objective of achieving a measurement of brain activity. The present work does not pretend to diagnose any disorder, only to measure and obtain results for its analysis from the engineering point of view but not the medical one. It was observed that the dolphin has a higher power of brain waves than human beings, but it is not a domesticated animal, requiring a period of learning and adaptation to the device presented in this work. This provoked some measurements to be incomplete.

The electrocardiograph device is designed to measure the heart rate for children with neurodevelopmental problems, using biomedical circuits connected wirelessly through the configuration of the Bluetooth communication protocol; Easy-to-handle water-pressure-resistant sheets are analyzed. Then, a *Sonoff*-type casing is designed, resulting in a series of portable, waterproof and wireless devices.

The device operation is validated by means of the Welch Power Spectrum Analysis, using distributed and parallel programming tools to measure cardiac activity obtained from samples with sampling rates: 30 (USB-Cam1${}_{S}$ and USB-Cam2${}_{S}$), 512 ( $EEG_{M}$ and $EEG_{S}$), 1000 ($ECG_{S}$) and 96,000 (USB-DAQ${}_{S}$) samples per second. Later, these samples are processed and analyzed to stabilize long-term results. The taking of the samples allows the processing of the data using mathematical and computational tools, and the processing indicates a correct device functioning, consistent with the expected results.

According to the above, the integrated system of Electroencephalographic, Electrocardiographic, Bioacoustic and Digital Image Acquisition Analysis provides neuroscience experts with tools to estimate the efficiency of a great variety of therapies. Our proposal is applied during a Dolphin-Assisted Therapy in a patient with Infantile Cerebral Palsy using the TAGM1, AD8232 BK type 3670 and Logitech-C920 sensors as well as the programming tools and toolboxes of *Matlab R2019b*, yielding the sampling stable and expected performance. Regarding the use and processing of the images recorded during therapy, these can be used to carry out an interactive video analysis of behavior or an analysis of micro-expressions. Hence, this system also provides work-tools to researchers in the field of artificial vision.

Furthermore, this system is tested for the acquisition of EEG signals in two more patients: (i) Intervention Patient with Obsessive–Compulsive Disorder (Patient 1) and (ii) Control Patient, i.e., without any neurodevelopmental problem (Patient 2), average of both patients is presented in bold. The results are presented three times (Table 2): (i) Before DAT, (ii) During DAT, and (iii) After DAT. The results regarding the Power Spectrum Density Analysis point out that both patients During DAT significantly increase their brain activity with respect to their activity at rest or Before DAT. In the case of Patient 1, a decrease in brain activity is observed with relaxation periods with respect to their activity Before DAT, indicating that DAT is used to regulate uncontrolled activity. However, for Patient 2, brain activity is increased During DAT, helping solve problems where attention processes are required.

On the other hand, from the outcomes yielded by the Self-Affine analysis, it is observed that in both patients negative correlations or antipersistent behavior are exhibited Before, During and After DAT. Moreover, in both patients During DAT also increases their brain activity with respect to their activity at rest or Before DAT, but not in a significant way. In addition, in Patient 1 DAT is more effective because the value of the Hurst exponent in this patient is greater than in Patient 2.

Finally, the present system is developed to acquire biomedical signals during a dolphin-assisted therapy, making it unique because the system has been developed for pediatric patients who can throw it away or get it wet. The quality and resistance of the materials are calculated for the possible treatment of a child with Autism Spectrum Disorder, since in most of them physical contact bothers them. In addition, the systems presented in the related works reviewed do not capture the signals from the biosensors because their proposal is that the EEG or ECG signals are processed in parallel but it does not ensure that the sample uses any synchronization technique during the acquisition of the samples. Therefore, the related works parallelize the digital signal processing, but do not parallelize the acquisition of biomedical signals, much less use spatio-temporal synchronization tools to form the most exact sampling times possible. The latter is necessary because brain activity can change in milliseconds, so ensuring the coincidence of the times is transcendental for health experts, doctors or psychologists. These professionals are provided with a scalable tool to develop experimentation ranging from detection of sleep to neurodevelopmental problems.

## Figures and Tables

**Figure 1 sensors-20-06991-f001:**
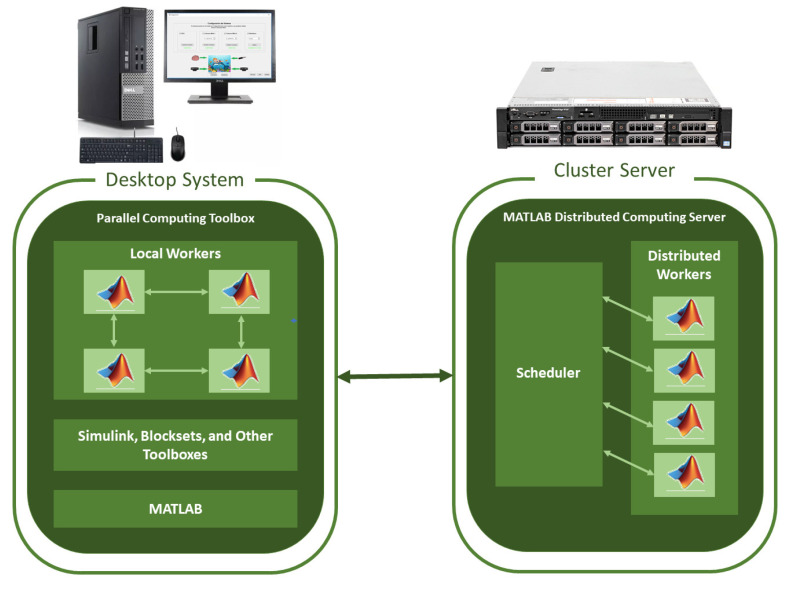
Matlab R2019b *Parallel-Computing* configuration for local and distributed architectures.

**Figure 2 sensors-20-06991-f002:**
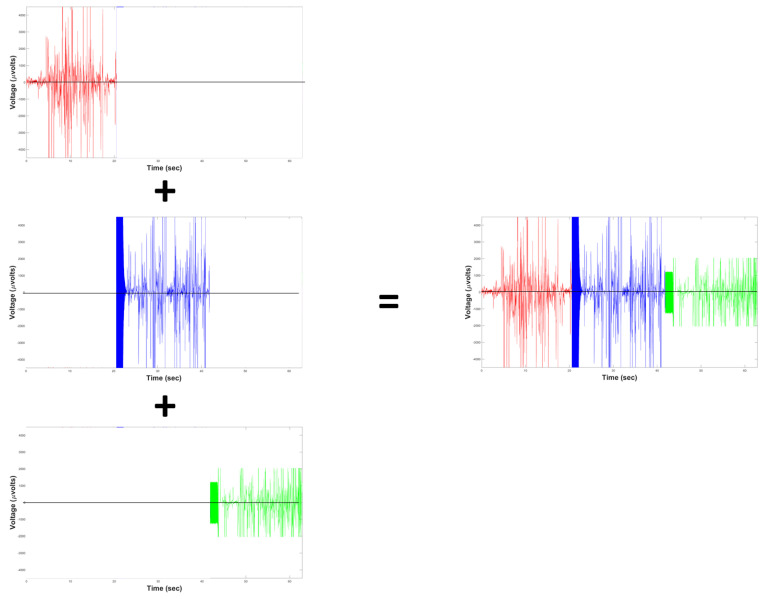
Sequential computing problem when EEG signals are acquired.

**Figure 3 sensors-20-06991-f003:**
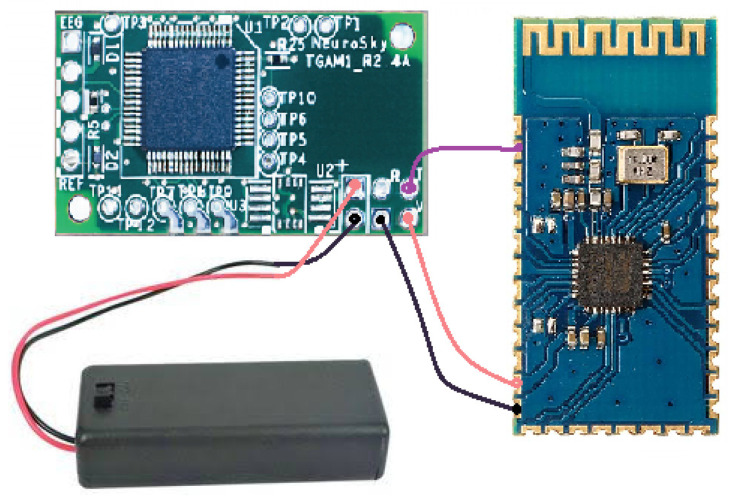
Electroencephalographic Biosensing System: ThinkGear ASIC Module 1 (TGAM1) and B26782H Bluetooth module.

**Figure 4 sensors-20-06991-f004:**
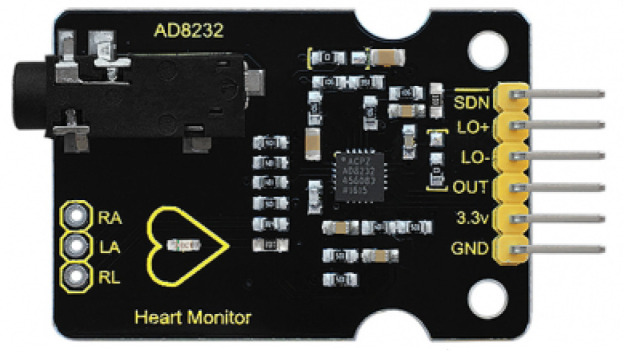
Electrocardiographic biosensor: AD8232.

**Figure 5 sensors-20-06991-f005:**
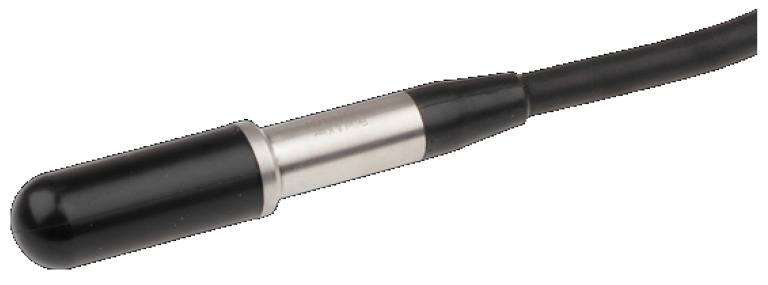
Bruel & Kjær Hydrophone Type 8103.

**Figure 6 sensors-20-06991-f006:**
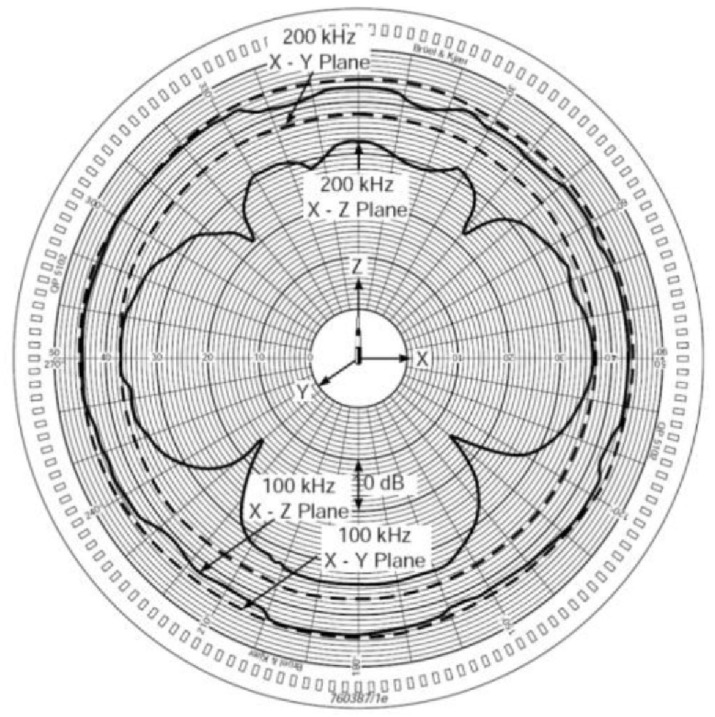
Bruel & Kjær Hydrophone Type 8103 directivity pattern.

**Figure 7 sensors-20-06991-f007:**
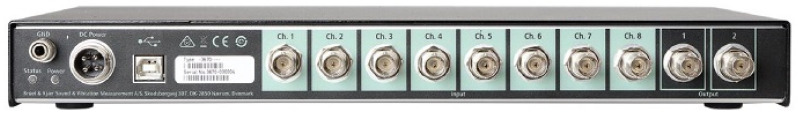
Bruel & Kjær Type 3670 Data Acquisition card.

**Figure 8 sensors-20-06991-f008:**
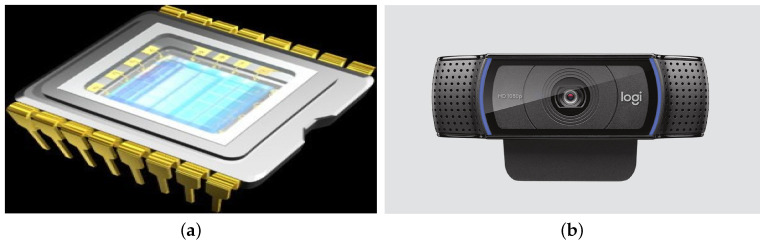
Image Acquisition System: (**a**) Charge-Coupled Device Sensor scheme and (**b**) Logitech C920.

**Figure 9 sensors-20-06991-f009:**
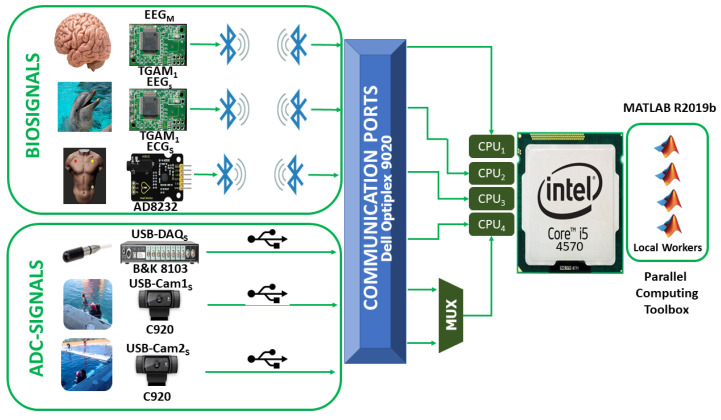
Architecture of System for monitoring biomedical signals using parallel computing.

**Figure 10 sensors-20-06991-f010:**
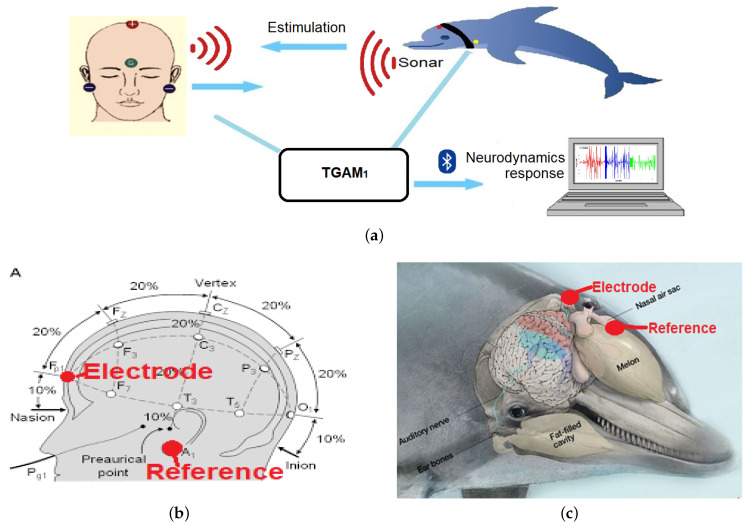
Design of the EEG interfaces: (**a**) general scheme. Placement of electroencephalographic systems: (**b**) $EEG_{M}$ and (**c**) $EEG_{S}$.

**Figure 11 sensors-20-06991-f011:**
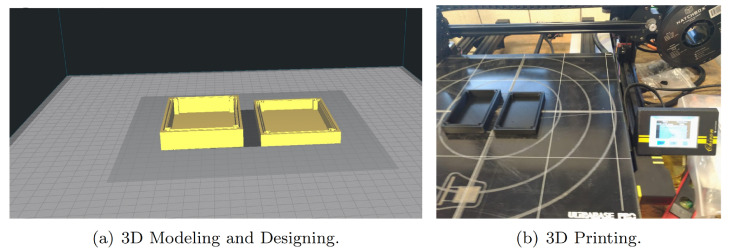
Designing of the waterproof case where TGAM1 Microcontroller is located.

**Figure 12 sensors-20-06991-f012:**
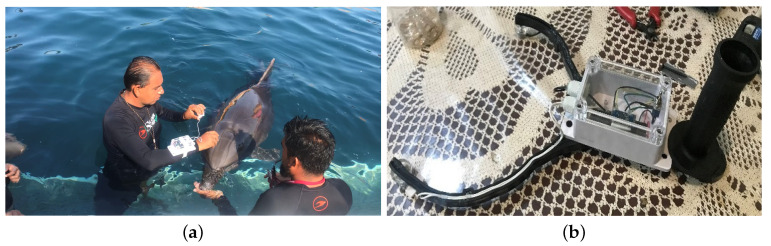
$EEG_{s}$ sensor design. (**a**) Searching for connection points and (**b**) Designing of an ergonomic handle.

**Figure 13 sensors-20-06991-f013:**
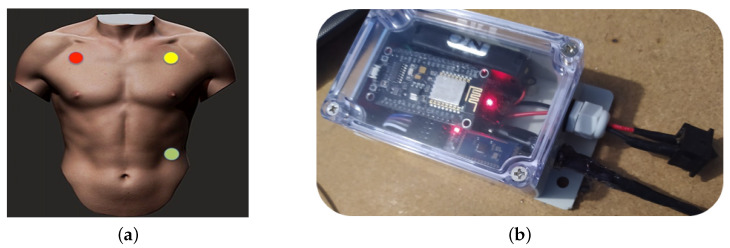
$EcG_{s}$ sensor design. (**a**) Searching for connection points and (**b**) Designing of an ergonomic handle.

**Figure 14 sensors-20-06991-f014:**
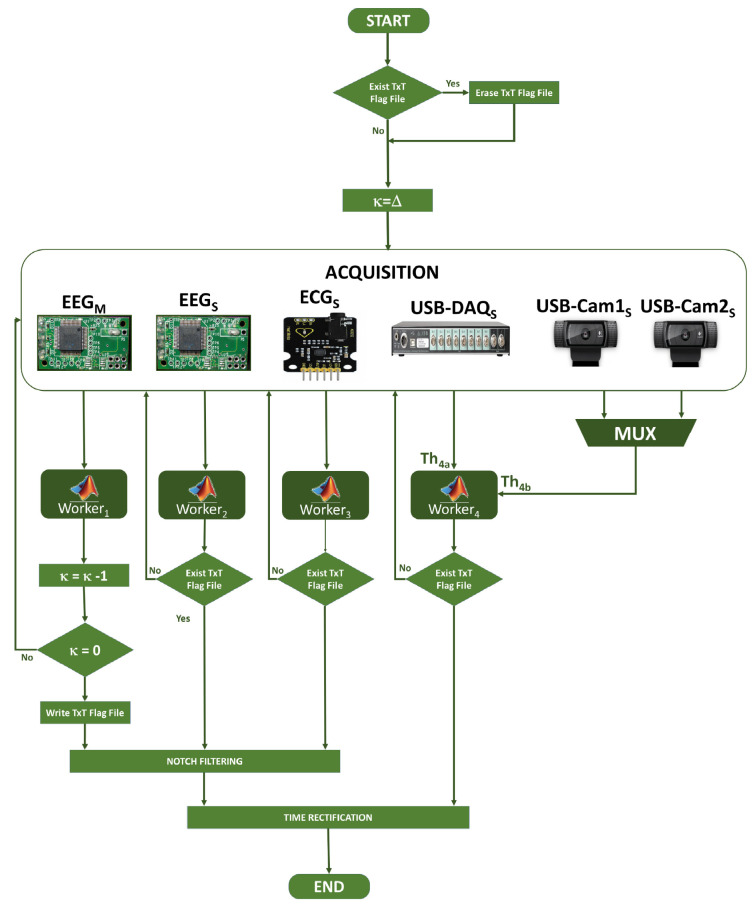
General Algorithm of System for monitoring biomedical signals using parallel computing.

**Figure 15 sensors-20-06991-f015:**
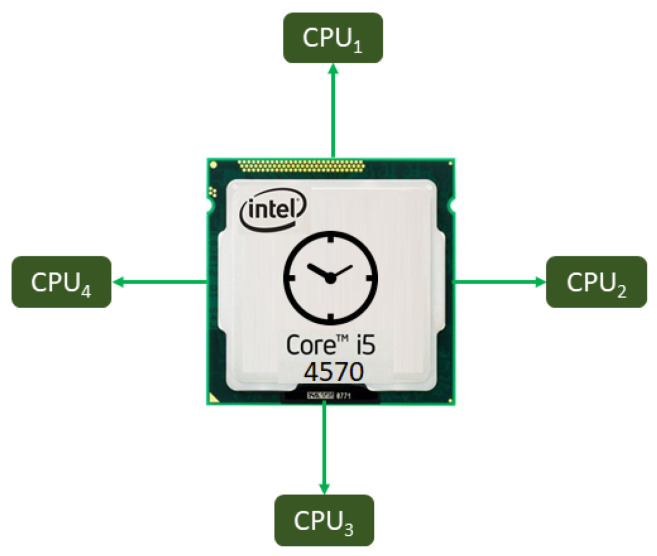
Time base of the microprocessor Intel i5 4570.

**Figure 16 sensors-20-06991-f016:**
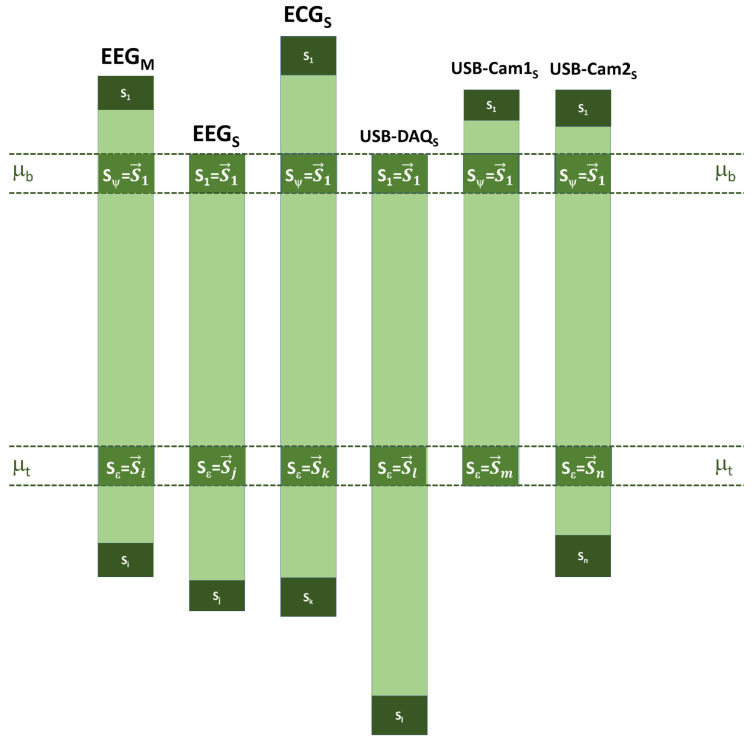
Synchronization step.

**Figure 17 sensors-20-06991-f017:**
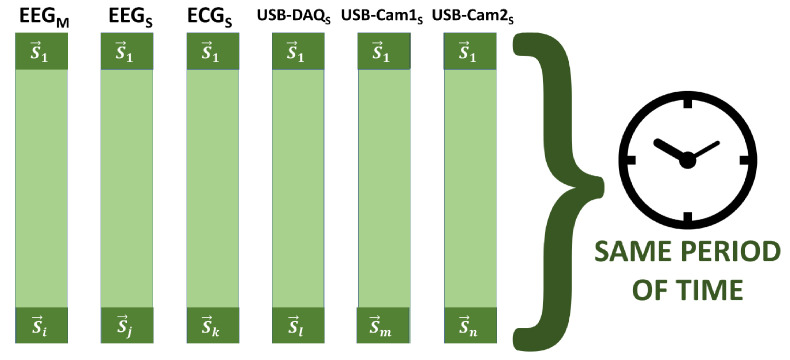
Sampling epoch.

**Figure 18 sensors-20-06991-f018:**
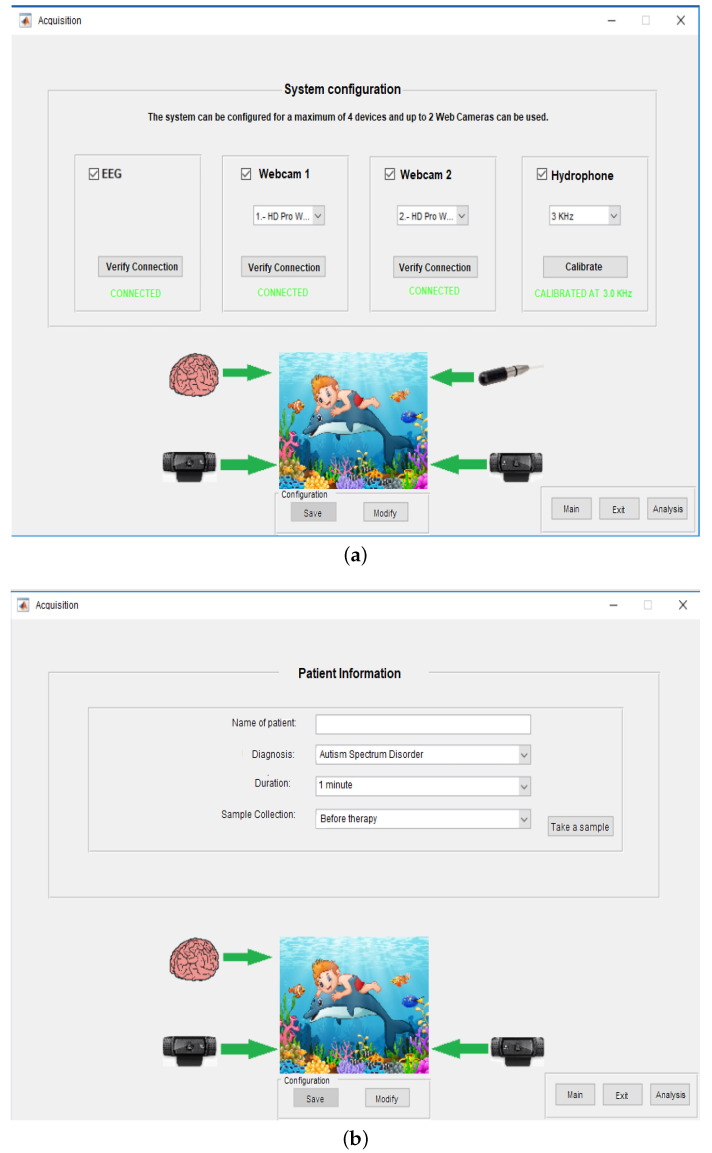
Integrated Development Environment: System for monitoring biomedical signals using parallel-computing IDE. (**a**) Windows of System Configuration and (**b**) Window of Patient Information.

**Figure 19 sensors-20-06991-f019:**
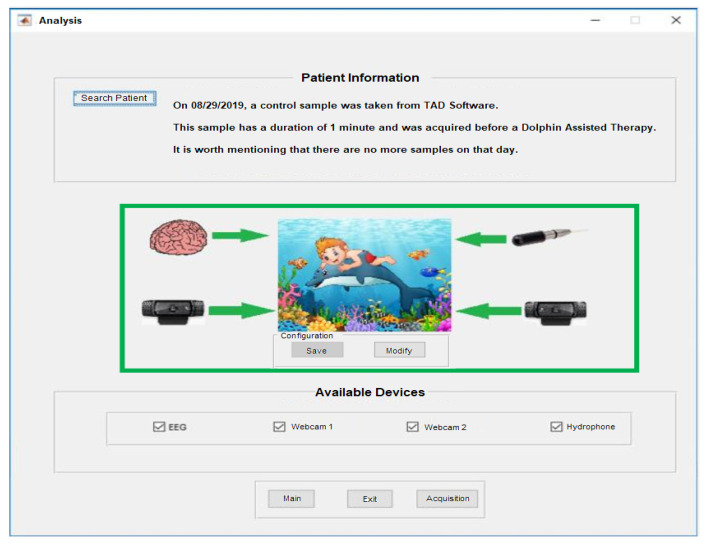
System configuration for the case study.

**Figure 20 sensors-20-06991-f020:**
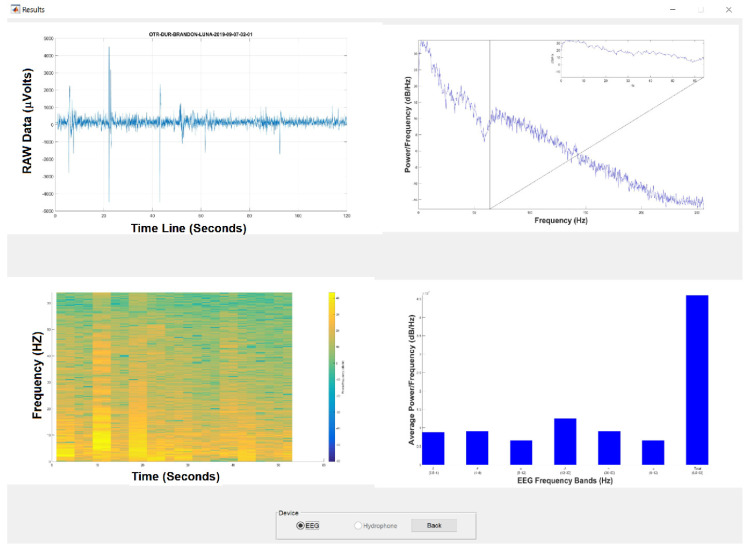
Results generated from the Analysis of a patient’s results for biomedical signals.

**Figure 21 sensors-20-06991-f021:**
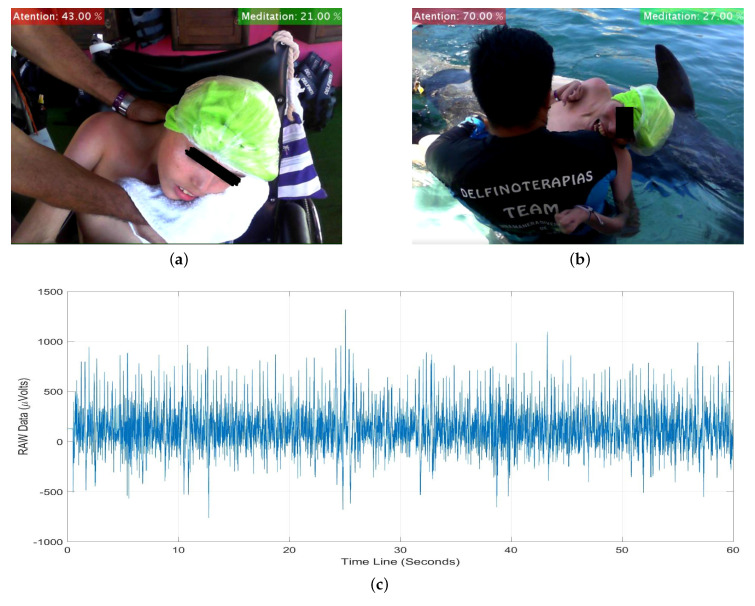
Acquisition of Time Series or RAW samples from $EEG_{M}$. (**a**) Before DAT, (**b**) during DAT, and (**c**) Example of generated time series.

**Figure 22 sensors-20-06991-f022:**
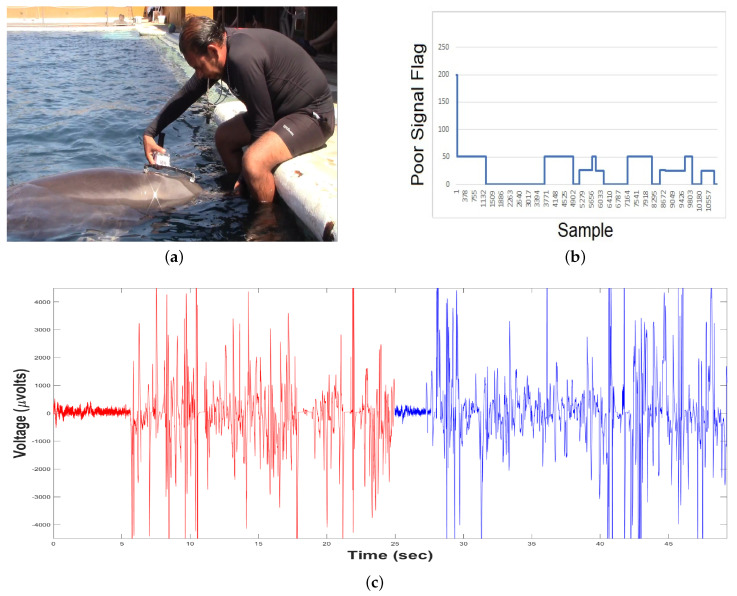
Original Time Series or RAW samples from $EEG_{S}$. (**a**) Dolphin Training, (**b**) POOR SIGNAL Flag, and (**c**) Time Series generated.

**Figure 23 sensors-20-06991-f023:**
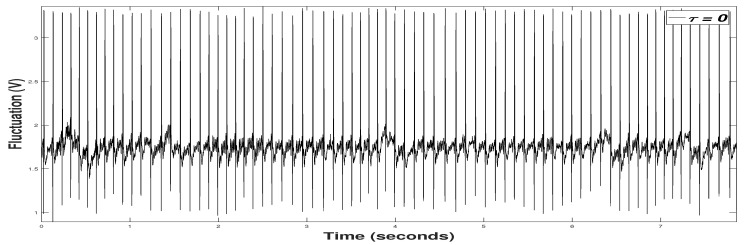
Original Time Series or RAW samples from $ECG_{S}$.

**Figure 24 sensors-20-06991-f024:**
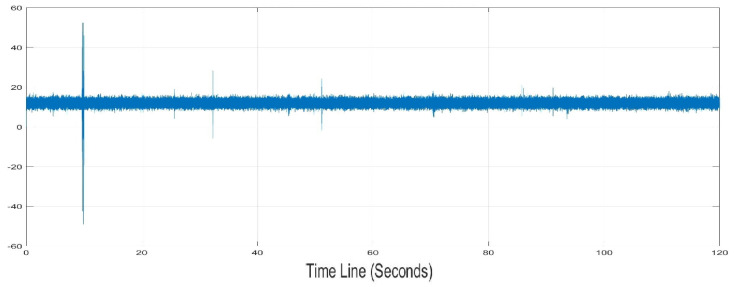
Original Time Series or RAW samples from USB-DAQ${}_{S}$.

**Figure 25 sensors-20-06991-f025:**
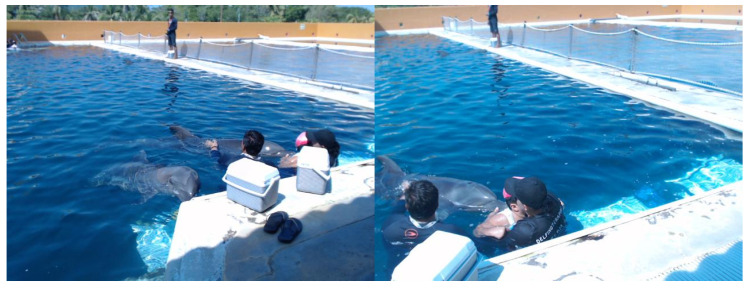
Original two-view images from USB-Cam1${}_{S}$ and USB-Cam2${}_{S}$.

**Figure 26 sensors-20-06991-f026:**
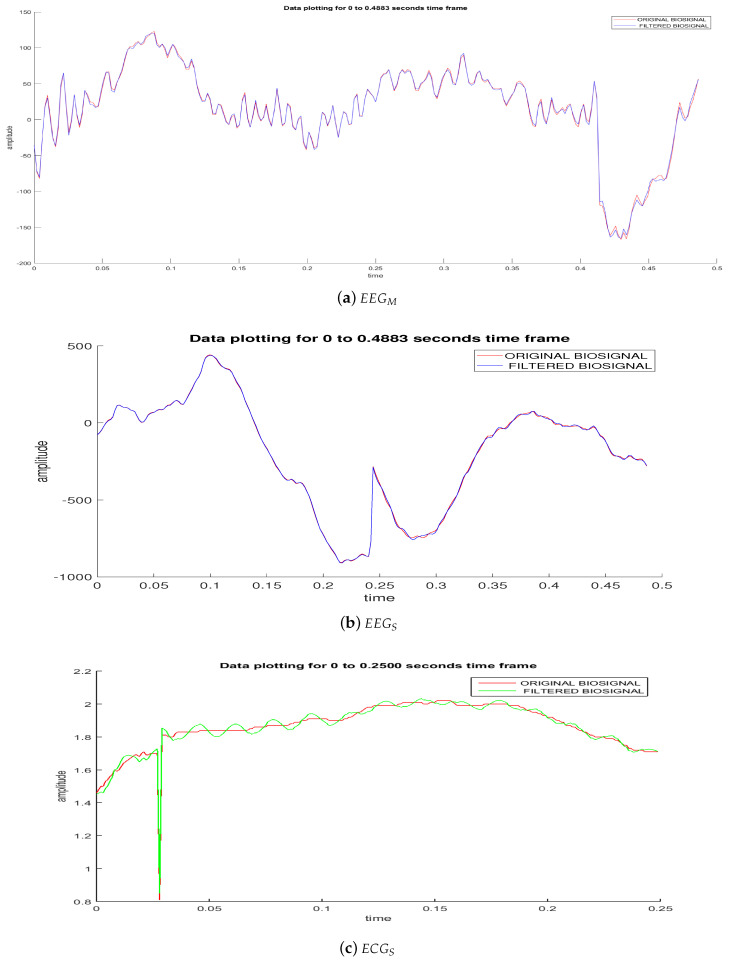
Analysis of biomedical signal interfaces: Notch Filtering.

**Figure 27 sensors-20-06991-f027:**
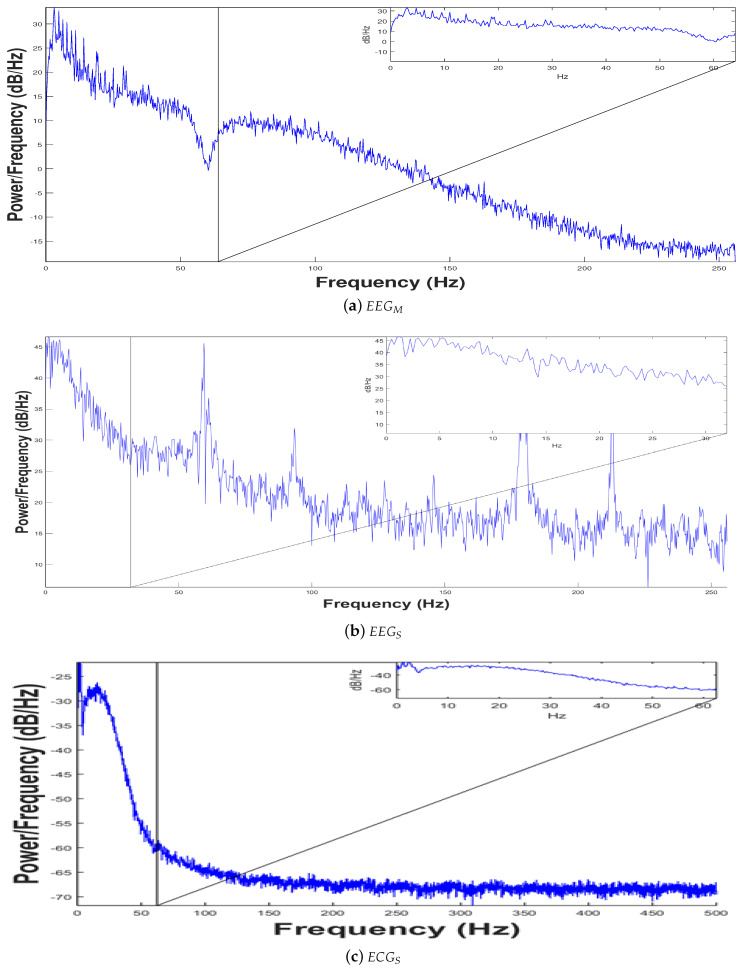
Analysis of biomedical signal interfaces: Power analysis or Periodogram.

**Figure 28 sensors-20-06991-f028:**
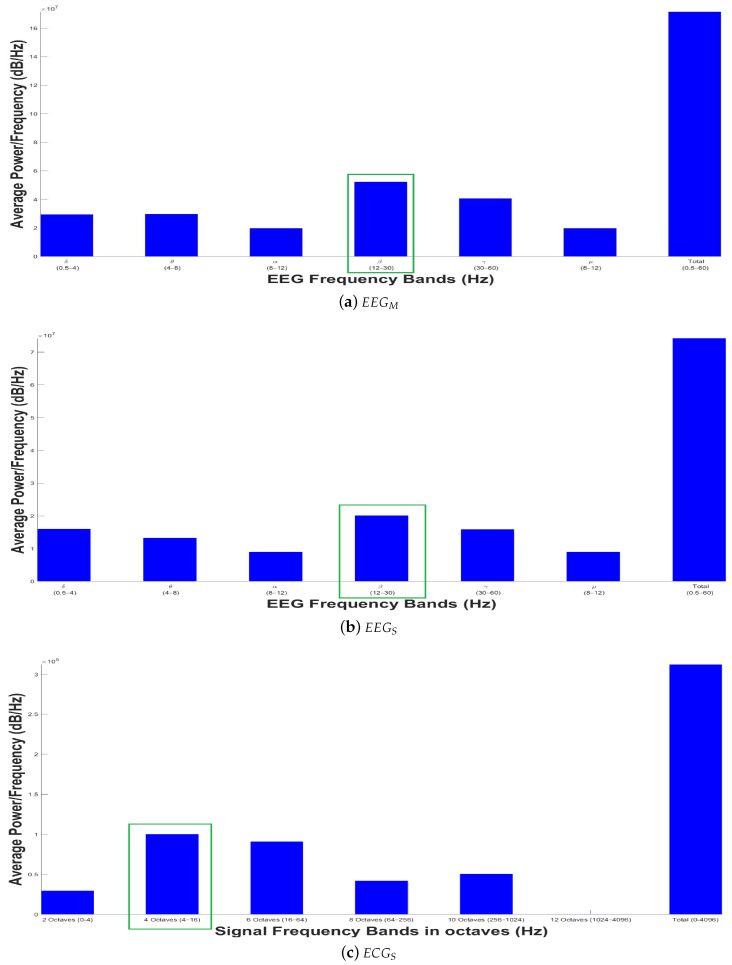
Analysis of biomedical signal interfaces: Average Spectrogram.

**Figure 29 sensors-20-06991-f029:**
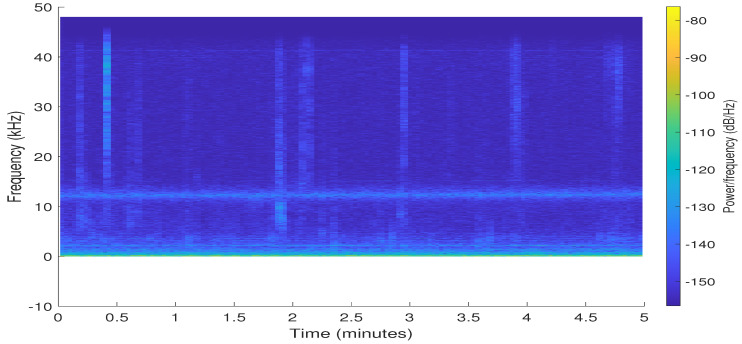
Analysis of USB-DAQ${}_{S}$ device: Spectrogram of time versus frequency of an underwater acoustic signal.

**Figure 30 sensors-20-06991-f030:**
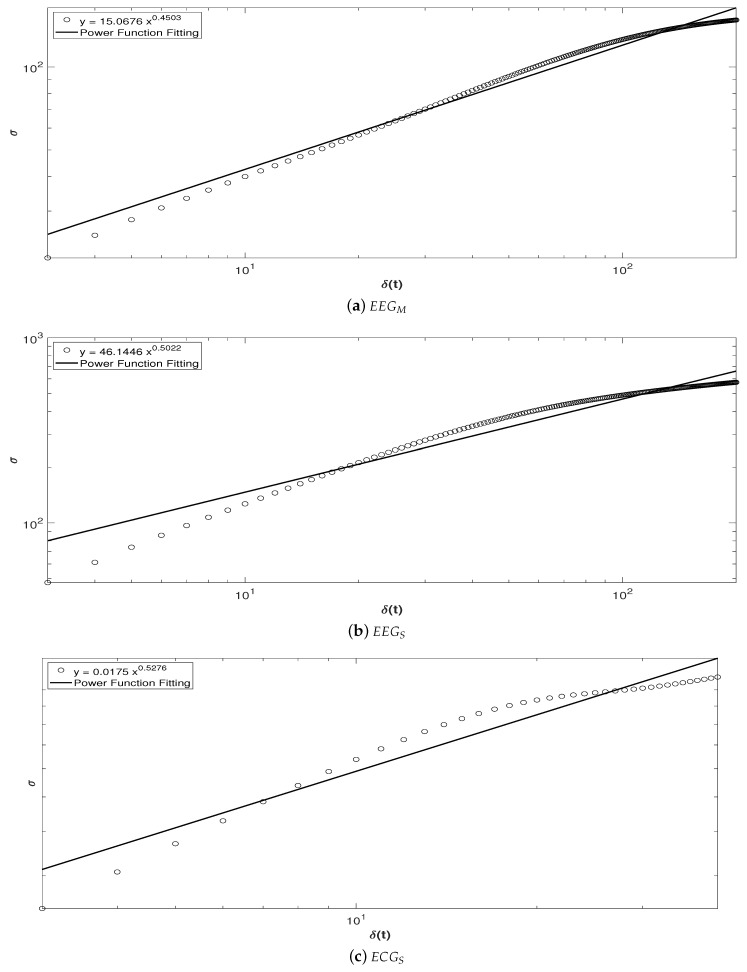
Self-Affine Analysis of: (**a**) $EEG_{M}$ in an intervention patient, (**b**) $EEG_{S}$ in a bottlenose dolphin, and (**c**) $ECG_{S}$ in an intervention patient.

**Table 1 sensors-20-06991-t001:** Comparison of the present proposal (in bold) with the existing works in the literature.

Feature	[9]	[10]	[14]	[15]	[16]	[17]	[18]	[19]	Proposal
**Article of**	Analysis	Analysis	Analysis	Design	Design	Analysis and Design	Analysis	Analysis and Design	**Analysis and Design**
**Signal Acquisition**	No	Yes	No	No	Yes	Yes	Yes	Yes	**Yes**
**Signal Processing**	Yes	Yes	Yes	No	No	Yes	Yes	Yes	**Yes**
**Sensors**	EEG	EEG	ECG	ECG	Camera	Camera	Bioacoustic, GPS	Bioacoustic, GPS	**EEG, ECG, Webcam, Bioacoustic**
**Number of channels**	6	14	2	1	1	1	4	3	**6**
**Epoch Synchronization**	No	No	No	No	No	No	No	No	**Yes**
**Analysis Tools**	Hilbert-Huang spectral entropy	Confusion matrix	Magnitud Response	Demonstration	Speedup Factor	None	Confusion matrix	PowerSpectrum Density	**PowerSpectrum Density and Fractal Geometry**
**Diagnosis**	No	No	Yes	No	No	No	No	Yes	**No**
**Experimentation**	Indoors	Indoors	Indoors	Indoors	Outdoors	Outdoors	Outdoors	Outdoors	**Indoors and Outdoors**
**Parallel Computing**	Local	Local	Distributed	Distributed	Local	Local	Distributed	Local	**Local**
**Parallel-Computing Usage**	Processing	Acquisition	Processing	Processing	Acquisition	Acquisition and Processing	Acquisition	Acquisition and Processing	**Acquisition and Processing**
**Artificial Intelligence Tool**	None	Neural Networks	None	None	None	SURF key points	Signal Classification	None	**None**
**Programming language**	Python	C/C++	Matlab	None	C/C++	C/C++	Python	Matlab	**Matlab**
**CPU Cores**	1	1	3	2	2	2	2	2	**4**

**Table 2 sensors-20-06991-t002:** Summary of the experimental results of Intervention Patient (Patient 1) and Control Patient (Patient 2), average of both patients is presented in bold.

	Self-Affine Analysis						Power Spectrum Density		
Patient	BEFORE		During DAT		AFTER		BEFORE	During DAT	AFTER
	*H*	CrossOver	*H*	CrossOver	*H*	CrossOver			
1	0.4405	119	0.4572	99	0.4235	136	1.74 $\times 10^{7}$	8.30 $\times 10^{7}$	1.41 $\times 10^{7}$
2	0.1978	33	0.2178	127	0.2151	185	1.53 $\times 10^{7}$	7.32 $\times 10^{7}$	2.35 $\times 10^{7}$
**Average**	**0.3192**	**76**	**0.3375**	**113**	**0.3193**	**161**	**1.64 $\times 10^{7}$**	**7.81 $\times 10^{7}$**	**1.88 $\times 10^{7}$**

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
