# Peer review of "Biomedical Signal Acquisition Using Sensors under the Paradigm of Parallel Computing"

_sensors, 2020, doi:10.3390/s20236991_

Round 1
Reviewer 1 Report
The main topic of this paper is the development of a complex system which can manage a number of bio-sensors in order to measure changes in different parameters and vital signs. The system is speed-up by using parallel computing techniques and the bio-signals can be interpreted after a processing stage.
In Introduction part, the state of art is according with the topic of the work, and the related works are well selected. It is strongly recommended that authors emphasize the contribution of this work compared with the most relevant papers on this topic. Also, Figure 2 needs a further explanation and better resolution, its hard to see the axes.
In Materials and Methods part, where the TGAM1 is explained, it appears “is the most used device…”, so please provide some bibliography supporting this. Figure 3 caption must be redefined in order to include the reference to Bluetooth module. Moreover, Figures 5, 6 and 7 present low resolution their improvement for readability is recommended. Also, the caption of Figure 8 must be more explanatory of two images, do they represent the same type of sensor? Additionally, more information about the electrode placement must be included, as scheme of EEG headband, polarity of ECG electrodes, etc. Furthermore, where the parallel computing instructions are presented, it is necessary to repeat the experiment several times in order to get more accurate results. Moreover, in Figure 20 it is difficult to see the details. Please review Figure 21 as far as the images do not match with the caption. In general, it is recommended to group the figures in order to ease the readability.
In Discussion part, it is recommended adding more parameters to enhance the ECG interpretability such as, heart rhythm variability measures. I strongly recommend changing the images from Figure 26 in order to clarify the filtering effects over signals, in (b) seems like calibration pulse is shown, and in (c) less than one beat is shown. Please, consider enhancing the image quality in Figure 27, they are difficult to see. In Figure 28 (c) the sampling frequency of ECG shows as 8 KHz, in line 424-426 appear “BIOSIGNALS interfaces, […], allowing to use a Notch frequency of 60/(512/2) for 3 dB of bandwidth." Please check if the sampling frequency is 8 KHz or 512 Hz. Consider enhancing the visibility of Figure 29 axes.
The Conclusion part is well explained and easy to follow.
In Bibliography, the references must be checked in order to harmonize the style
Please check the whole document grammar and typos.
Author Response
"Please see the attachment."

Reviewer 2 Report
Dear authors,
I found the work very interesting but the manuscript is not correctly organized and it is too long. Additionally, there exists a self-plagiarism problem since there are too many parts of the work already published (see references 40 and 41).
The manuscript must be rewritten in order to be eligible for publication. You can find some recommendations bellow:
- The Abstract must be more focused, and it must contain, at least, the main objective, the case study and the main results of the work.
- The Introduction is too long. Not in content but in extension. Also, the Objective of the work is not correctly defined (lines 158-159) but it is in the Discussion (636-639).
- Materials & Methods:
- In overall, this section is too long. It must be just a description of the tools used, there is too much literature therein.
- There is no Case Study definition, only lines 294-296.
- There is no a high-level diagram of the programming.
- Matlab code should be in a Supplementary Materials.
- Line 469 should be the Case Study.
- Line 49-50 quotes “these signals are synchronized to place them at the same sampling moment and are analyzed using a Power Spectral Density and Fractal Geometry” but there is no any Result regarding the Fractal or Power Spectral Density analyses.
- Linked with previous, in Signal Processing section there is no any signal processing algorithm explained.
- Line 540 “real-time analysis”, has to be justified.
In Discussion section, line 621, you must compare your results with other works’ results. No your Mat&Methods section with other works’ Mat&Methods sections.
Author Response
"Please see the attachment."

Reviewer 3 Report
The manuscript describes the design and implementation of a wireless medical system for acquiring and processing medical biosignals by using parallel computing architectures.
The application of the developed system in the article is quite interesting but the contribution added by the article are difficult to find due to the following reasons.
Minor comments:
The abstract does not reflect the content of the article, a brief description of the materials and methods used, the technical results obtained after the system testing stage and few conclusions and further improvements.
The article is divided into inappropriate parts: an introduction with a very long state of the art, also a very long materials and methods section containing a lot of details about the background knowledge of EEG, ECG, hydrophone, DAQ, and USB cameras, many of them being well-known, a results section with big pictures and charts. This makes the article difficult to follow and understand.
Major comments:
The proposed system contains a mixture of medical sensors and video cameras with individual raw data processing. This makes the entire system questionable from parallel processing point of view. Instead of using medical physiological parameters extracted from the acquired biosignals and using for classification or information fusion, as examples, by using custom developed parallel processing algorithms, simple processing methods are applied for each biosignal/image. The simply use of an integrated toolbox in Matlab for parallel processing, without rigorous explanations about the needs if using it, cannot be considered a true contribution added by the article.
But I think the main lack of the article consists in insufficient evaluation of the proposed system, either from technical and/or medical points of view, or in comparison with other similar medical monitoring systems.
Author Response
"Please see the attachment."

Round 2
Reviewer 1 Report
The changes made to this paper are improved their quality, but some issues must be to consider. First, figures 5 and 6 still present low quality. Second, Figure 7 (b) is different sized to Figure 7 (a) and its quality can be improved. Third, Figure 20 axes are too small to observe them. Fourth, Figure 26 (b) shows a calibration pulse, and it is not correct to represent the filtering stage effect on signals, also the axes labels are difficult to read. Finally, Figure 29 still present axis to small to correct visualization.
Author Response
"Please see the attachment."

Reviewer 2 Report
Congrats to the authors, the manuscript has been overally improved and it is ready for publication.
Author Response
"Please see the attachment."

Reviewer 3 Report
After the addressing most of the comments and remarks the article is now in a better condition.
In my opinion the main lack of the article remains the insufficient evaluation of the proposed system, either from technical and/or medical points of view, or in comparison with other similar medical monitoring systems.
Fortunately, the main key point of the article by using dolphin-assisted therapy with the proposed medical system may be interesting to some of the journal's users.
Author Response
"Please see the attachment."
